# Physiological Approaches Targeting Cellular and Mitochondrial Pathways Underlying Adipose Organ Senescence

**DOI:** 10.3390/ijms241411676

**Published:** 2023-07-19

**Authors:** Pieter de Lange, Assunta Lombardi, Elena Silvestri, Federica Cioffi, Antonia Giacco, Stefania Iervolino, Giuseppe Petito, Rosalba Senese, Antonia Lanni, Maria Moreno

**Affiliations:** 1Dipartimento di Scienze e Tecnologie Ambientali, Biologiche e Farmaceutiche, Università degli Studi della Campania “Luigi Vanvitelli”, Via Vivaldi 43, 81130 Caserta, Italy; pieter.delange@unicampania.it (P.d.L.); giuseppe.petito@unicampania.it (G.P.); rosalba.senese@unicampania.it (R.S.); antonia.lanni@unicampania.it (A.L.); 2Dipartimento di Biologia, Università degli Studi di Napoli “Federico II”, Monte Sant’Angelo, Via Cinthia 4, 80126 Naples, Italy; assunta.lombardi@unina.it; 3Dipartimento di Scienze e Tecnologie, Università degli Studi del Sannio, via De Sanctis snc, 82100 Benevento, Italy; silvestri@unisannio.it (E.S.); fecioffi@unisannio.it (F.C.); agiacco@unisannio.it (A.G.); stefaniaiervolino15@gmail.com (S.I.)

**Keywords:** adipose tissue, senescence, mitochondria, metabolic disorder, aging, diet, caloric restriction, exercise

## Abstract

The adipose organ is involved in many metabolic functions, ranging from the production of endocrine factors to the regulation of thermogenic processes. Aging is a natural process that affects the physiology of the adipose organ, leading to metabolic disorders, thus strongly impacting healthy aging. Cellular senescence modifies many functional aspects of adipose tissue, leading to metabolic alterations through defective adipogenesis, inflammation, and aberrant adipocytokine production, and in turn, it triggers systemic inflammation and senescence, as well as insulin resistance in metabolically active tissues, leading to premature declined physiological features. In the various aging fat depots, senescence involves a multiplicity of cell types, including mature adipocytes and immune, endothelial, and progenitor cells that are aging, highlighting their involvement in the loss of metabolic flexibility, one of the common features of aging-related metabolic disorders. Since mitochondrial stress represents a key trigger of cellular senescence, and senescence leads to the accumulation of abnormal mitochondria with impaired dynamics and hindered homeostasis, this review focuses on the beneficial potential of targeting mitochondria, so that strategies can be developed to manage adipose tissue senescence for the treatment of age-related metabolic disorders.

## 1. Introduction

The intensive aging of the population is generally accompanied by metabolic derangement-based diseases with an impaired quality of life, therefore representing the main cause of human disease and death worldwide. One of the most vulnerable organs in aging is the adipose organ, with alterations in various biological and physiological processes that, in turn, affect the organism’s general well-being [1]. The adipose organ is well known for its function in both energy storage and mobilization in response to nutrient availability and body needs and in thermogenesis; thus, it controls the energetic balance of the organism. In a recent study, researchers found that the first age-related immune response was detectable in white adipose depots [2] featuring alterations in molecular and physiological processes affecting its metabolic flexibility. A dysfunctional adipose organ in aging promotes chronic low-grade inflammation, deregulation of insulin action, and lipid infiltration in the elderly, therefore representing an attractive target for intervention in aging-related metabolic disorders [3,4]. The two principal mammalian adipose tissues are white (WAT) and brown (BAT) adipose tissue (Figure 1). White adipocytes represent many cells of WAT depots and present a single lipid droplet and a low number of mitochondria; brown adipocytes, which, in rodents, are predominantly present in the interscapular region, contain multilocular lipid droplets and a markedly higher number of mitochondria when compared with white adipocytes. A third subtype named brown-in-white (brite) or beige adipocytes, interspersed within the WAT of rodents and humans, can be induced to browning, a process that switches from a white-adipocyte-like to a brown adipocyte-like phenotype [5]. 

WAT, abundantly distributed in the body [6], is primarily involved in modulating energy homeostasis in response to changes in systemic energy levels and plays a role in endocrine signaling in multiple metabolic responses [7]. Indeed, WAT secretes lipokines, adipokines, and exosome microRNAs, which influence a variety of physiological processes [8]. Although BAT accounts for only a small proportion of the total amount of adipose tissue, its ability to induce adaptive thermogenesis has a fundamental metabolic impact [9], and emerging evidence suggests that BAT is also involved in endocrine signaling by secreting a variety of signaling molecules [10]. These metabolic/thermogenic responses are mediated by the characteristics of the mitochondrial population. Indeed, the mitochondria of white and brown adipocytes have been functionally linked to the maintenance of intracellular metabolism with brown adipocyte mitochondria highly expressing uncoupling protein-1 (UCP1) that enables energy dissipation in thermogenic/metabolic adaptative responses [11]. In response to specific stimuli, the induction of beige adipocytes in white adipose tissue, also known as WAT browning or beiging, improves glucose and lipid metabolism. Beige adipocytes express UCP1 and, like brown adipocytes, are highly metabolically and thermogenically active, thus adding to the role of BAT in body temperature and energy homeostasis, and body weight control [12,13].

Notably, all types of adipose tissue undergo finely balanced structural and metabolic remodeling in response to physiological stimuli. This plasticity ensures metabolic flexibility to meet the body’s needs and involves dynamic communication of adipocytes with a variety of progenitor cells, immune cells, vascular cells, and sympathetic neurons, which, in turn, respond to upstream innervation and blood supply [14]. Mitochondrial dysfunction disturbs adipocyte metabolic flexibility, leading to metabolic complications, such as insulin resistance, obesity, and type 2 diabetes. Therefore, understanding the molecular mechanisms underlying adipocyte mitochondrial dysfunction could provide potential opportunities for the prevention or delaying of full-blown metabolic disturbances by therapeutic intervention [15]. Limitation/loss of adipose tissue plasticity causes reduced or dysfunctional responses to physiological cues, and age-related adipose tissue senescence has emerged as an important factor in the development of comorbidities associated with metabolic disorders [16]. Therefore, interventions that delay or prevent adipose tissue senescence have been shown to have the potential to prolong health and lifespan [17]. Here, we mainly focus on the related cellular and molecular biological mechanisms underlying age-related effects on adipose tissue leading to dysmetabolism. This review is not intended to exhaust the topic but to provide more information to uncover the potential of cellular senescence mechanisms as a therapeutic strategy mainly targeting mitochondria for future anti-senescence interventions. 

## 2. Age-Related Adipose Tissue Dysfunction

During aging, the adipose organ alters its mass and is redistributed throughout the body, which is accompanied by an increased risk of metabolic-associated diseases, including insulin resistance, cardiovascular diseases, and diabetes [18]. Aging is associated with a preferential increase in fat accumulation in visceral adipose tissue (VAT) and a decrease in subcutaneous fat (SAT), partially due to defects on SAT adipogenesis related to hyperactivated inflammatory pathways [19,20]. SAT and VAT differentially affect metabolism, with SAT being considered beneficial for metabolism, whereas VAT expansion is considered detrimental [21]. Thus, the age-related redistribution of adipose tissue in favor of VAT depots expansion is diagnostic for metabolic dysfunctions of aging. Indeed, fat redistribution during aging is correlated with an increased risk of metabolic abnormalities, particularly insulin resistance, resulting in increased risk of cardiovascular disease and type 2 diabetes [21,22]. A significant aging-related reduction in BAT mass and activity and in beige adipocytes’ formation has been detected; thus, a decreased adipose thermogenic/metabolic activity significantly adds to adipose organ dysfunction and increased visceral body fat accumulation [23]. 

Aging is associated with decreased UCP1 activity in BAT, and the related mitochondrial dysfunction is recognized to play an important role in the pathogenesis of several age-related metabolic disorders [24,25]. Moreover, UCP1 gene expression is suppressed by increased levels of proinflammatory cytokines inducing a repression of BAT thermogenesis in the aging process [26]. One molecular player contributing to the age-associated drop in thermogenesis is the winged helix factor forkhead box protein A3 (Foxa3), which is involved in BAT mass reduction and related age-associated metabolic dysfunction [27]. During aging, elevated levels of Foxa3 in BAT lead to the suppression of peroxisome proliferator activated receptor γ coactivator 1 α (PGC-1α) levels through interference with the cAMP responsive element binding protein 1 (CREB)-mediated induction of the PGC-1α promoter [27]. Accordingly, Foxa3-null mice have increased BAT, enhanced browning, and thermogenic capacity upon aging, accompanied by a less expanded adipose tissue, decreased insulin resistance, and increased longevity, further supporting a role for Foxa3 in energy expenditure and in age-associated metabolic disorders [27]. Aging adipose tissue is characterized by a reduced browning capacity associated with the age-related reduction in Sirtuin 1 (SIRT1), a histone deacetylase which induces a brown phenotype in white adipocytes by deacetylating peroxisome-proliferator-activated receptor gamma (PPARγ) [28]. The functional plasticity of adipose tissue during the aging process is also impaired by the decline of the proliferative and differentiative capacity of adipose progenitor and stem cells (APSCs), thus rendering it less able to adequately store lipids, resulting in the exposure of the surrounding tissues to increased amounts of free fatty acids exceeding physiological needs, leading to lipotoxicity [29]. This phenomenon is a critical mechanism of the metabolic syndrome and has a strong negative impact on the health of the elderly [30]. The impaired differentiative capacity of preadipocytes during aging is linked to altered levels of adipogenic factors, partly because of increased levels of anti-adipogenic factors [31,32]. In addition to these transcription factors, microRNAs (miRNAs) and short non-coding RNAs of 17–20 nucleotides also participate in the dysfunction of preadipocytes in the aging process through the regulation of mRNA expression and translation in adipocyte production pathways. Age-associated impairments in preadipocyte differentiation capacity have been found to correlate with miR-143, which promotes adipocyte differentiation via the extracellular-signal-regulated kinase (5 ERK5)-PPARγ pathway [33], as extensively reviewed elsewhere [34].

The structure and function of thermogenic adipose tissue change over the course of life [35]. The recent characterization of brown and beige thermogenic adipocytes in adult humans and the definition of their role in whole-body energy homeostasis have increased interest in their biology, and both rodent models and human research have highlighted age-related changes in thermogenic adipocytes [36]. An age-related accumulation of lipids leads to a reduction in mitochondria and UCP1 content and the ability to recruit beige adipocytes [37,38], resulting in a decrease in the thermogenic potential of adipose tissue, which may be mechanistically linked to the increased risk of age-associated metabolic alterations. Aging human and murine beige progenitor cells have been shown to exhibit a cellular senescence phenotype, which explains their age-dependent decline [39]. When in young beige progenitors the senescence pathway is genetically or pharmacologically induced, there is an arrest of their potential for cold-induced beige adipocyte generation [39]. In contrast, genetic or pharmacological reversal of cellular senescence signaling by targeting the p38-mitogen-activated protein kinases (p38/MAPKs) pathway, an upstream regulator of the Ink4a/Arf pathway, in aged mouse or human beige progenitor cells rejuvenates cold-induced beige adipocyte formation [39]. The inhibition of p38/MAPK via SB202190 (SB) in aged mice leads to a cold-exposure-induced increase in beige adipocyte formation, which appears to underlie the increase in glucose uptake and reduction in adiposity [39]. Furthermore, treatment with SB restores the beige adipocyte formation potential of aged human vascular stromal cells [39]. Altogether, these results indicate that anti-aging or senescence modalities could be a strategy to induce beiging, thus improving metabolic health in aging. In this context, it is of importance to underline the recently opened new perspective on the importance of the metabolic characteristics of “physiologically humanized mice” in addressing the question of thermogenic adipose tissue similarities in mice and humans [40]. As described in detail in the symposium report by Cannon et al. [41], studies on mice conducted under “physiologically humanized” conditions (middle-aged mice fed a high-fat diet and housed in thermoneutral conditions, 30 °C) compared to “standard” conditions (young mice fed a chow diet and housed at standard temperature, 20 °C) indicate that classic BAT exhibits the molecular markers of human thermogenic adipose tissue (see [41] and references therein). In particular, human BAT and classical BAT from humanized mice share expression patterns of particular genes that have been suggested to be “markers” for brown versus brite/beige adipose tissues (i.e., suggested “brown” marker genes—F-Box Protein 31 (Fbxo31), V-like antigen 1/myelin protein zero-like 2 (Eva1/Mpzl2), and early B-cell factor-3 (Ebf3); and brite/beige—cidea, T-box 1 (Tbx1), and Transmembrane Protein 26 (Tmem26); see [41] and references therein), therefore supporting the idea of mouse BAT being more appropriate for translational studies with respect to the brite/beige counterpart.

## 3. Adipose Tissue Inflammaging and Cell Senescence

Aging is associated with circulating inflammatory markers independent of body mass index [42]. Aged adipose tissue is strongly characterized by a complex interplay between general cellular senescence, immunosenescence, and inflammation in determining poor metabolic health outcomes in old age involving various cellular and molecular mechanisms. As fully explained in the following section (Section 4), among the several players associated with adipose tissue inflammaging and cellular senescence, one key player is the mitochondrion. Mitochondrial function is involved in cellular senescence, chronic inflammation, and age-dependent stem cell activity. Soro-Arnaiz et al. [43] found that the aging of white adipocytes is associated with an inefficiency of mitochondrial complex IV, due to the HIF1A-induced repression of subunit Cox5b. Silencing Cox5b promotes derangements in the young mouse’s white adipocyte, while its recovery prevents age-related oxidative damage [43].

Aging-associated immunosenescence leads to chronic low-grade inflammation, termed “inflammaging”. Inflammation of adipose tissue involves the release of pro-inflammatory cytokines such as TNF-alpha, interleukin (IL)-1β, IL-6, and monocyte chemoattractant protein (MCP-1/CCL2) by adipose tissue macrophages, and these factors are also associated with the presence of senescent cells expressing the biomarker p16INK4A (a tumor suppressor and cell-cycle regulator), leading to immunosenescence [44,45,46]. Indeed, immunosenescence is intricately linked with inflammaging [47], leading to the accumulation of senescent/dead cells, adipocyte hypertrophy, and, hence, increased plasma levels of free fatty acids and lipopolysaccharides. Chemokines-induced recruitment of immune cells to adipose tissue impairs adipose tissue plasticity during aging (for a review, see [48]), with the activation of the nuclear factor kappa-light-chain-enhancer of the activated B cells’ (NF*k*B) signaling pathway promoting a pro-inflammatory bias in adipose tissue macrophages [49].

Age-related senescence of adipose tissue involves an irreversible cell-cycle arrest, followed by the appearance of a senescence-associated secretory phenotype (SASP) consisting of cytokines, chemokines, growth factors, proteases, Activin A, and miRNA [50]. In many age-related metabolic alterations, the secretion of SASP factors derived from adipose senescent cells, reactive oxygen species (ROS) production, infiltration of proinflammatory immune cells into adipose tissue, and increased DNA damage has been reported to impair metabolic adipose organ functions by causing cell death in combination with a state of local and systemic hyperinflammation (Figure 2) [51,52]. 

Senescent progenitor cells or preadipocytes are less able to differentiate into mature white, beige, or brown adipocytes [53,54,55]. Furthermore, senescent adipose progenitor cells are involved in the failure of the age-related beiging process [56]. It has been shown that Activin A, which is known to negatively affect adipogenesis when secreted by human senescent preadipocytes, renders non-senescent-adipose-tissue-derived progenitor cells (APC) senescent and suppresses their adipogenic capacity [57]. Activin A inhibition in differentiated progenitors exposed to senescent cells enhances the expression pattern of key adipogenic factors [57]. Indeed, by inhibiting Janus Kinase (JAK) in the senescent cells of 18-month-old mice, Activin A was blunted, while adipogenesis was increased [56]. Human SAT biopsies from individuals with hypertrophic obesity display senescent preadipocytes undergoing poor differentiation, characterized by increased activation of p53 and p16ink4, which, in turn, reduce the levels of the zinc finger protein ZNF521, leading to hypertrophic expansion of white adipocytes [51]. p53 plays a central role in the induction of cellular senescence, as well as in the regulation of adipogenesis [56]. Downregulation of p53 is fundamental to allow the adipogenic program to progress [56]; therefore, its activation in senescent adipocytes provides one of the reasons for the impaired adipogenesis [51]. Overall, senescence associated with p53 activation inhibits adipocyte differentiation, blocks insulin-dependent glucose transport, and increases lipid release, thus exacerbating the inflammatory state and non-response to insulin [58,59]. As with aging, p53 levels also increase in WAT in type 2 diabetes, and its overexpression in rodent WAT leads to systemic insulin resistance, thus supporting the association between increased p53 levels in WAT and metabolic disorders in obesity and senescence, also related to age [60]. 

The accumulation of senescent cells may also result in the decline in stemness and adipogenesis of aged APSCs [52]. Indeed, studies have found that senescent human adipocyte progenitors, via the paracrine pathway, inhibit the adipogenesis of surrounding non-senescent progenitors [52]. Not more than 20% of adipocyte progenitors accumulate lipids when cocultured with senescent cells, while lipid accumulation when cocultured with non-senescent cells occurs in over 50% of progenitors.

These effects are likely to be related to Activin A, IL-6, tumor necrosis factor alpha (TNF-α), interferon-γ (IFN-γ) as senescent adipose progenitor, and/or other senescent SASP cell components [52]. The accumulating senescent cells exceed the quantity that can be effectively removed and release many chemokines that, in turn, disturb the immune cell response [52]. The secretion of SASP-induced guanosine-5′-triphosphate (GTP)-binding RAS-like 3 (DIRAS3) markers in human preadipocytes leads to chemokine and inflammatory cytokine secretion, which is associated with the metabolic dysfunctions in adipose tissue [61]. Dysregulated levels of proinflammatory cytokines promote immune cell infiltration and increase local and systemic inflammation [62]. Indeed, coculture of 3T3-L1 preadipocytes with a macrophage cell line has been shown to induce preadipocyte TNF-α secretion, supporting the existence of a bidirectional interplay between adipocytes and immune cells that exacerbates the so-called “inflammaging” of senescent adipose tissue [63]. Coculture experiments also demonstrated that senescent cells influence the function of neighboring non-senescent cells through Activin A and TNF-α [64,65], leading to a decline in adipose tissue plasticity [63]. Furthermore, the secretion of factors such as IL-6, Activin A, and TNF-α by senescent cells contributes to insulin resistance in metabolic tissues, thus impeding adipogenesis and attracting immune cells, events that make senescent cells contribute to the risk of type 2 diabetes and metabolic syndrome [65].

## 4. Mitochondrial Dysfunction in Adipocyte Senescence

Mitochondria have a central role in the cell’s metabolic processes, from ATP and energy production to epigenetic regulation [66]. The white, brown, and beige adipocyte subtypes have distinct mitochondrial abundance, function, and plasticity [67]. WAT mitochondria trigger adipocyte differentiation, lipogenesis, lipolysis, and fatty acid oxidation [67]. BAT contains relatively more mitochondria with considerable thermogenic power [68,69,70,71,72]. High-resolution quantitative mass spectrometry allowed for a more direct and accurate comparison between the in vivo mitochondrial proteomes of WAT and BAT adipocytes [73]. The significant qualitative and quantitative differences in mitochondrial proteomes between WAT and BAT adipocytes indicated that brown adipose mitochondria share a higher similarity with those of skeletal muscle, being enriched in proteins of pathways related to fatty acid metabolism, the TCA cycle, and oxidative phosphorylation activity (OXPHOS) other than UCP1 [73]. Beige adipocytes have mitochondria similar to those of BAT adipocytes since they contain UCP1 and are able to utilize both lipids and carbohydrates as substrates [73]. Although some of their actions are adipocyte-type-specific, adipose mitochondria, in general, are essential for maintaining tissue function in metabolic homeostasis, interacting with other metabolically active tissues. Systemic metabolism is regulated by brown and beige adipocytes through signaling pathways that are functionally associated with skeletal muscle metabolism and systemic energy expenditure. Recently the small molecules 3-methyl-2-oxovaleric acid (MOVA), 5-oxoproline, and β-hydroxyisobutyric acid (BHIBA), identified by using metabolomic approaches, were shown to be synthesized in browning adipocytes and secreted via monocarboxylate transporters. These molecules, termed metabokines, induce the beiging of white adipocytes, as well as skeletal myocyte mitochondrial oxidative energy metabolism both in vitro and in vivo [74]. In humans, metabokine concentrations in plasma and adipose tissue correlate with markers of adipose browning and are inversely associated with body mass index [74]. MOVA and 5OP induce fatty acid oxidation via cyclic AMP–protein kinase A–p38 MAPK (cAMP–PKA–p38 MAPK) signaling [74]. BHIBA, by acting through the mammalian target of rapamycin (mTOR), regulates metabolism-related gene expression in adipocytes and myocytes and induces β-oxidation. In addition to metabokine, BHIBA has also been shown to be released from skeletal myocytes in response to the overexpression of PGC-1α, a master regulator of muscle metabolism, playing a key role in the adaptive response of muscle to exercise [75]. Such a physiological exchange of metabokines between beige and brown adipocytes and muscle results in reduced adiposity, increased energy expenditure, and improved glucose and insulin homeostasis in obese and diabetic rodents [74]. 

Mitochondria play regulatory roles in cellular senescence and modulate the SASP (Figure 3) [76]. Key features of mitochondrial dysfunction during cellular senescence involve a reduced respiratory capacity, a decreased membrane potential, and increased ROS production, as well as altered mechanisms involved in the homeostasis of mitochondrial mass and morphology [76]. Mitochondrial dynamics, mainly through the crucial processes of fusion and fission, maintain functional mitochondria and mass homeostasis. Mitochondria in senescent cells increase significantly in size and mass [77,78], and the latter is due to altered mitophagy/degradation processes [79,80], leading to Senescence-Associated Mitochondrial Dysfunction (SAMD). The accumulation of dysfunctional mitochondria and altered mitochondrial dynamics in cellular senescence are closely intertwined in the induction of senescence-associated pathways [81,82]. While mitochondrial turnover decreases in senescent cells, increased mitochondrial biogenesis leads to enhanced respiratory activity and ROS production [83]. The induced oxidative stress leads to cell damage, cell-cycle arrest, and senescence [84,85,86]. Senescent adipose mitochondrial dysfunction is an essential trigger of the induction of the complete senescent phenotype, including a proinflammatory SAPS [87,88,89]. Thus, mitochondrial dysfunction can both induce and be a consequence of cellular senescence and appears to maintain the senescent phenotype in adipose tissue profoundly affecting cellular bioenergetics (e.g., alteration in electron transport chain (ETC), interference in tricarboxylic acid (TCA) cycle, and mitochondrial DNA (mtDNA) depletion) [87]. In addition to the already mentioned intramitochondrial factors, cytosolic and extracellular factors also determine the maintenance of adipose mitochondrial health. These factors, which are involved in cellular senescence, fall into the categories of epigenetic factors [90,91], miRNAs [92,93] and long noncoding RNAs [94,95], hormones [96,97], and inflammatory molecules [98,99,100]. Adipocytes produce extracellular vesicles (EVs) that, by facilitating intercellular and interorgan crosstalk, regulate metabolism in and beyond adipose tissue [101,102]. Adipocyte EVs have been shown to contain active, and functional mitochondria have been shown to modify recipient cells’ metabolic activity [103,104]. Thus, it is perhaps not surprising that senescent adipocytes have been shown to release EVs that exert negative effects on surrounding cells and tissues [105]. 

It has been reported that white adipocytes transfer their mitochondria to macrophages, resulting in a transcriptionally distinct macrophage subpopulation [101]. Genome-wide CRISPRCas9 knockout screening revealed that the uptake of mitochondria depends on heparan sulfate (HS). Having lower HS levels on WAT macrophages, high-fat-diet (HFD)-induced obese mice exhibit decreased intercellular mitochondrial transfer from adipocytes to macrophages. This impaired uptake of mitochondria promotes the accumulation of fat mass and reduces systemic energy expenditure without affecting calorie intake or physical activity [101]. Related mitochondrial-based therapy strategies offer a potentially new therapeutic approach for the treatment of metabolic disorders [105].

## 5. Adipose–Skeletal-Muscle Crosstalk in Aging

The age-related metabolic inflexibility of adipose tissue not only impacts factors regulating intercellular communications but also those involved in inter-organ communication, thus leading to systemic effects that contribute to functional decline and metabolic disorders. The contribution of lipids released by adipose tissue to age-related skeletal muscle quality and function is relevant [106]. Lipids and their derivatives accumulate both within and between muscle cells (myosteatosis), leading to mitochondrial dysfunction, disturbing fatty acid oxidation, and enhancing ROS production, thus determining local lipotoxicity, which, along with being associated with functional decline of skeletal muscle, also induces insulin resistance and inflammation [106,107,108,109]. It has been reported that the secretion-associated senescent phenotype reveals reduced expression of contractile proteins and increased insulin resistance in human muscle cells [110]. In vitro exposure of human muscle cells to the secretome of senescent human adipose tissue reduces myosin II heavy chain and troponin expression [111]. Furthermore, it has been reported that the obese visceral adipose secretome modulates myotubular expression of the contractile proteins myogenic differentiation factor (MyoD) and myogenin and decreases the levels of growth factors such as insulin-like growth factor 2 (IGF2) and insulin-like growth factor-binding protein 5 (IGFBP-5), as well as titin levels [111], and this, in turn, impacts sarcomere structure and elasticity, and neuromuscular function. Indeed, in elderly humans, senescent adipocytes and immune cells infiltrating skeletal muscle exert detrimental effects on muscle fibers [112]. Interestingly, transplantation of senescent preadipocytes impairs grip strength and overall physical function in young mice [113]. These observations imply that adipose tissue impacts muscle function during aging. However, it is necessary to consider that skeletal muscle also acts on adipose function. Skeletal muscle secretes myokines that balance glucose and lipid metabolism in the entire body and also regulate adipose tissue metabolism with varying effects [114]. Myokines, including irisin, meteorin-like, myostatin, b-aminoisobutyric acid, fibroblast growth factor 21 (FGF-21), and myonectin, regulate the beiging of adipose tissue. Myonectin functions to decrease plasma-free fatty acids by increasing CD36 and FATP1 in adipose tissue, which stimulates adipose free fatty acid uptake; for a review, see [115] and the references therein. During aging, distinct myokines, including irisin, TNF-α, IL-6, IL-10, IL-1b, FGF21, and myostatin, negatively impact adipose tissue, promoting an inflammatory feedback loop that triggers the release of adipose cytokines (TNF-α, IL-1β, and IL-6) by adipose tissue macrophages, with immunosenescence and impaired body metabolism as a consequence [116,117]. 

Skeletal muscle loses mass and function during aging, a process termed sarcopenia. Of the factors that affect muscle aging at the cellular and molecular level, synergizing with each other, ROS play a primary role. Recently, several studies have shown that muscle stem cells are involved in the progression of sarcopenia as their number and activity regresses. Sarcopenia features altered protein synthesis and mitochondrial dysfunction. Aging induces several characteristic changes in skeletal muscle mitochondria, including increased deletions in mtDNA [118] and increased oxidative stress [119]. In addition, aging is associated with a diminished ability to induce muscle biogenesis in response to stimuli, including muscle contraction [120]. Aging further features decreased mitochondrial content, decreased resting and maximal oxygen (O2) consumption, and decreased activity of the TCA-cycle and OXPHOS enzymes [121]. These disrupted mechanisms form the basis of the decline in muscle function in relation to the secretome of the adipose tissue mentioned above, with age. Mitochondria can regulate the myonuclear domain of senescent muscle cells and the plasticity of skeletal myofibers under physiological and pathological conditions, indicating that these organelles are crucial for the maintenance of muscle cell activity [122]. Therefore, the process of mitochondrial decay is an established biomarker of aging and is also considered the main driving force of skeletal muscle aging.

## 6. Targeting Adipose Senescence to Improve Metabolic Function

Cellular senescence leads to metabolic dysfunction in adipose tissue, muscles, the liver, and the pancreas, resulting in obesity and type 2 diabetes [123]. The preadipocytes of obese subjects show increased senescence markers, including P53, IL6, C–C motif chemokine ligand 2 (CCL2), Cyclin Dependent Kinase Inhibitor 1A (CDKN1A), and Cyclin Dependent Kinase Inhibitor 2A (CDKN2A) [124], and limited replicative potential, resulting in reduced adipogenesis and increased adipose cell size compared to those of lean subjects [125]. In human SAT, senescence markers, including P53, galactosidase beta 1 (β-Gal), and serpin family E member 1 (SERPINE1), were increased in hypertrophic obesity, and this increase was boosted further in similarly obese type 2 diabetic individuals [125]. Mature adipose senescent cells display increased levels of various senescence markers. These include β-Gal, a hydrolase enzyme for which increased expression and activity in senescent cells indicates increased lysosomal mass, p16, for which the increase during arrested cell cycle is one of the hallmarks of cellular senescence, plasminogen activator inhibitor 1 (PAI-1, encoded by SERPINE1), a potent and rapid-acting inhibitor of both of the mammalian plasminogen activators involved in the regulation of cellular senescence via phosphatidylinositol 3-kinase (PI3K)–protein kinase B (PKB)–glycogen synthase kinase 3 (GSK3)–cyclin D1 pathway, p53, and the phosphorylation of mitogen-activated protein kinase 8 (also known as JNK1) [125]. These events corresponded to decreased levels of adipogenic markers, PPARγ and glucose transporter 4 (GLUT4), and serine 473 phosphorylation of PKB in type 2 diabetes compared to lean individuals [125]. In obese people, the senescence of subcutaneous adipocytes rather than omental adipocytes has been shown to correlate with hyperinsulinemia [126]. Prolonged hyperinsulinemia induces cyclin-D1-mediated cell-cycle progression in mature adipocytes, progressively leading to the activation of senescence pathways [126]. 

The development of technologies for the analysis of genome-wide DNA methylation (DNAm) levels allowed for the characterization of age-related methylation patterns in adipocytes [127]. Various genes involved in adipocyte differentiation, lipid metabolism, and inflammation show altered DNA methylation patterns in metabolic disorders [128]. A newly discovered methylase enzyme, METTL4, responsible for the methylation of N6-methyladenine (6 ma), was identified as a promotor of adipogenesis in 3T3-L1 cells [129].

Overall, this evidence reveals that senescent adipose tissue triggers the development of metabolic dysfunction. In human non-differentiated and differentiated adipose cells, the chemotherapeutic agent Doxorubicin induces senescence by increasing p21, Zinc Finger Matrin-Type 3 (ZMAT3), and cyclin D1. Simultaneously, the adipogenic markers adiponectin, PPARγ, and fatty acid binding protein 4 (FABP4) were diagnostic for cellular de-differentiation [125]. Accordingly, improved indices of adipose function are achieved upon the removal of senescent cells, which include the reduction in adipocyte size, improvement in adipogenic potential, and decrease in macrophage infiltration [130,131].

The activation of BAT mitochondria has been reported to improve dyslipidemia and insulin resistance; thus, BAT thermogenesis is emerging as a strategy to overcome these disturbances [132,133,134]. More recently, WAT beiging has been found to result in improved body energy metabolism and resistance to diet-induced obesity [135,136]. Notably, beige adipocytes exist in humans [137], and their activity can be induced by several factors, including hormones and physical activity that enhances metabolic processes [138]. In recent years, general interest in the scientific community has increased concerning elucidating pharmacological and non-pharmacological strategies to preserve adipose plasticity, to prolong longevity, and to prevent metabolic diseases by targeting senescence. Based on physiological processes, several strategies, including senolytic and senostatic/senomorphic approaches, as well as non-pharmacological interventions, have been devised to target senescent cells and positively impact obesity-related adipose dysfunction in metabolic diseases; for a review, see [139,140]. Treatments that improve mitochondrial function reduced SASP and increased healthy aging. Lifestyle factors targeting mitochondria may have an impact on the induction and evolution of cellular senescence (Figure 4).

Therapeutic approaches for cellular senescence:Caloric restriction

As an anti-senescence approach, caloric restriction (CR), increasing mitochondrial biogenesis and bioenergetic activity through SIRT1 activation [141], still represents the gold standard. The CR-mediated beneficial effect on mitochondrial activity has been correlated with reduced senescent cell burden and SASP, reduced oxidative stress, and increased autophagy [142], all of which are involved in senescent metabolic derailment. CR can protect cellular senescence by interfering with the damage’s source; decreasing, for example, oxidative stress and inflammation; or increasing autophagy, thus repairing/eliminating already present damage [142]. It is well established that sirtuins are critical in reducing oxidative stress. Mitochondrial dysfunction caused by SIRT3 downregulation can induce senescence. CR induces SIRT3 expression [143], and also SIRT1 was reported to mediate aspects of the CR response [144] by reducing levels of oxidative stress within the cell [145]. By activating WAT AMP-activated protein kinase (AMPK), CR induces appropriate levels of autophagy to remove defective organelles and proteins from the cytoplasm [146]. Indeed, CR-mediated activation of the AMPK/SIRT1/PGC-1α energy-sensing network and mitochondria-mediated thermogenesis may ameliorate obesity and associated metabolic disorders [146]. Interestingly, the surgical removal of VAT in rats induces the effect of CR on longevity by about 20% by enhancing insulin action [147]. Corrales et al. [148] reported that long-term (7–12 months) CR attenuates the decline in SAT function with age and decreases the extent of fibro-inflammation. Furthermore, CR has been shown to promote SAT browning, hence identifying the browning process in response to food restriction, providing a strategy to prevent the adverse metabolic effects in middle-aged animals. Spadaro et al. [149] recently showed that about 14% CR for 2 years in healthy humans induces the transcriptional reprogramming of pathways regulating bioenergetics and the anti-inflammatory response in adipose tissue. Expression of the platelet-activating factor acetylhydrolase (PLA2G7)-coding gene was decreased in humans undergoing CR, while inactivation of the same gene in aging mice decreased inflammation and improved several metabolic functions in [149]. These data reveal adipose PLA2G7 repression as one of the potential mechanisms associated with the beneficial effects of CR.

CR increases circulating ketone bodies, especially beta hydroxy butyrate (BHB), which has been demonstrated to have anti-aging effects [150]. Accordingly, aged mice fed a ketone ester diet display increased circulating BHB levels, an improvement in the competency of aged progenitors for adipogenic differentiation, and restored beige fat activity [151].

The CR mimetics metformin and rapamycin have similar beneficial potential, but effective doses exceed those achievable by in vivo therapeutic intervention without incurring adverse side effects [152]. Dietary CR mimetics (e.g., non-starch polysaccharides, polyphenols, and other phytochemicals) have better safety profiles and future studies should focus on their effects [153]. Indeed, polyphenols (i.e., resveratrol, quercetin, and hydroxytyrosol) have been shown to improve insulin sensitivity by modulating adipocyte mitochondrial function through activating the SIRT1-PGC1α axis [154]. Recently, it has been shown that polyphenols induce energy metabolism by activating thermogenesis in BAT [155]. Moreover, α-linoleic acid, a plant-derived polyunsaturated fatty acid, promotes mitochondrial density and fatty acid oxidation, exerting an anti-obesity effect in humans and in rodents [156]. Considering the wide range of dietary compounds and targets that can produce CR-like effects, the identification of new and safe CR mimetics that positively impact senescence is within reach. 

Physical Exercise

Another recognized senostatic strategy with minimal side effects is exercise. Exercise ameliorates adipose cellular senescence, dysregulation of adipokines, and adipocyte bioenergetics in a weight-loss-independent manner and promotes the recovery of glucose homeostasis in obesity [157,158]. It has been shown that a 12-week exercise program reduces circulating senescence biomarkers in older adults [158], and a recent study in humans suggests that the senescent adipose cell number is inversely related to physical function in elderly women [159]. Exercise exerts a beneficial effect on high-fat-diet-induced adipose senescence by preventing the accumulation of senescent cells and significantly reducing markers including p16INK4 and SA, thus almost abolishing the negative metabolic effects of high-fat diets [157]. Therefore, therapeutic approaches to control or suppress SASP, such as physical exercise, may offer a strategy to counteract the impact of senescent cells on the genesis of metabolic dysfunction. Exercise-mediated targeting of the regulatory systems of SASP, NF-kB [160], and mTOR [161] could provide a promising means to prevent the secretion of pro-inflammatory cytokines in senescent cells [162]. 

Physical exercise also improves the health of aging skeletal muscle [163,164]. Ample evidence suggests that exercise reverses age-related changes by stimulating myokine production, by mediating communication between muscle and surrounding organs, and by promoting muscle health. In addition, exercise modifies epigenetic markers through ROS, regulates mitochondrial-homeostasis-related proteins, and maintains or improves skeletal muscle mitochondrial health, as extensively reviewed in [165]. Moreover, miRNAs are emerging as possible candidates to orchestrate adipose metabolic adaptations. miRNA levels are decreased in the adipose tissue of aged [166] and obese [167] mice, caused by the downregulation of the type III endoribonuclease DICER. DICER is a key enzyme for the biogenesis of most miRNAs in adipocytes [167]. Exercise training increases mouse and human adipose DICER levels [168]. In mouse skeletal muscle and adipocytes, DICER upregulation depends on AMPK signaling, resulting in an increase in adipose miRNA expression, increasing physical performance, as well as whole-body and adipose metabolic fitness, features that are lost in DICER KO mice [168]. One such miRNA, miR-203-3p, limits glycolysis in adipocytes [168]. Both miR-203-3p and DICER induce BAT differentiation and thermogenesis [168]. Thus, DICER integrates signals from exercising skeletal muscle in adipose tissue and thus is a good candidate for promoting metabolic flexibility in the organism.

Intermittent fasting

Intermittent fasting (IF) is gaining popularity because of its beneficial effects on overall health [169]. IF has beneficial effects on adipose tissue, as it promotes its remodeling, reduces markers of inflammation and inflammasome signaling components [170], promotes mitochondrial fusion, and improves mitochondrial function in the white visceral adipose tissue of HFD mice [171]. After two months of treatment, IF (24 h fast on three alternating days weekly) improves glucose tolerance and insulin resistance in HFD-fed mice and reduces adipocyte hypertrophy and markers of inflammation. These include the infiltration of macrophages and signaling components of the NLRP3 inflammasome [172]. Moreover, the abovementioned, as well as a complete alternate-day fasting protocol, induces increased energy expenditure and UCP1 expression in WAT in HFD-fed mice and reduces inflammatory markers [173,174]. In obese human subjects, various related fasting protocols, such as intermittent continuous energy restriction for 2 days a week and alternate-day fasting, reduce total fat mass [175] and circulating markers of inflammation [176].

Fasting-induced SIRT1 activation increases NAD+ levels, enhances mitochondrial biogenesis, and delays senescence [177]. SIRT1 is known to increase mitochondrial metabolism and reduce ROS [178]. SIRT1 activity suppresses adipogenesis and induces lipolysis by modifying PPARγ activity and activating the AMPK pathway, and in addition, it represses inflammation by repressing NF-κB, NLR Family Pyrin-Domain-Containing 3 (NLRP3), and mTOR pathways. Moreover, SIRT1 regulates extracellular matrix (ECM) deposition and fibrosis [179]. Different IF regimens have been shown to strongly induce WAT browning and increase the expression of mitochondrial UCP1 in rodents, possibly linked to SIRT1, resulting in an ameliorated metabolic state [180,181]. AMPK is also related to mitochondrial function. Fasting-mediated AMPK activity governs adipose mitochondrial metabolism and homeostasis. Chronic activation of AMPK controls mitochondrial network dynamics and mitochondrial interaction with other organelles, while increasing fatty acid oxidation [182]. Both fasting and exercise activate AMPK in adipose tissue, leptin, and adiponectin, hypoglycemic drugs are activators of adipose AMPK. Increased AMPK activity correlates with the effect of IF on the enhanced adipose adiponectin levels even in the presence of HFD [183]. Therefore, AMPK is a strategic target for maintaining mitochondrial function following CR, IF, and exercise to preserve adipose senescence and prevent metabolic diseases. Further studies are warranted to evaluate the interactions between CR and CR mimetics and should focus on other anti-aging interventions, such as exercise and intermittent fasting, to provide a broader basis for pharmacological and nonpharmacological therapies/strategies. This includes combinations of these interventions, some of which have indeed been shown to be systemically beneficial [184], synergistically improving functional features of adipose tissue [184]. Indeed, very recently, the combination of energy restriction and exercise has resulted in a significant reduction of visceral fat mass and an increased glycosylated hemoglobin in obese women [185] and improved body composition in healthy males [186].

## 7. From Evidence That Adipose Senescence Drives Age-Related Metabolic Diseases to Approaches to Identify Senolytics

Senescent adipocyte precursors, when transplanted into young animals, induce a phenotype reminiscent to that of aging, accelerating the onset of age-related diseases and premature death [187]. Adipose progenitor cells form the principal target of senolytics in fat. Indeed, treatment with the senolytic cocktail dasatinib plus quercetin (D+Q) alleviates metabolic dysfunction both in genetically or diet-induced obese mice and in mice with transplanted senescent preadipocytes [113]. Xu et al. [113] studied surgically excised omental adipose tissue explants from obese individuals (BMI 45.5 ± 9.1 kg/m^2^; age 45.7 ± 8.3 years) and found that these indeed contained naturally occurring senescent cells. These authors reported that treatment with D+Q (1 μM + 20 μM) vs. vehicle (V) for 48 h decreases the senescence markers p16INK4A and SA-Gal β, correlating with a selective decrease of naturally occurring senescent cells and of the cells’ secretion of pro-inflammatory cytokines. Moreover, intermittent oral administration of senolytics to mice that either underwent transplantation of senescent cells or were naturally aged alleviated physical dysfunction, while increasing survival by 36% [113].

Adipose tissue from mice treated with D+Q (5 mg/kg and 50 mg/kg) or the vehicle for various periods shows decreased p16INK4A, centromere binding protein (CENBP)-, and p21Cip1-expressing senescent cells [187]. Importantly, the effects of intermittent D+Q administration are consistent with reduced senescent cell burden, rather than effects unrelated to senescence [187], given that the elimination half-life of each drug is <12 h [188]. This suggests a putative therapeutic strategy that, if further confirmed in pre-clinical studies, should be considered for clinical trials. Apart from reducing adipocyte hypertrophy, senolytics increase the subcutaneous-to-visceral-adipose ratio, improve glucose tolerance and insulin sensitivity, and reduce circulating inflammatory mediators and promoting adipogenesis in obese mice [189]. Furthermore, senescent cell elimination prevents the accumulation of transplanted monocytes and macrophages into intra-abdominal adipose tissue [189]. 

A bioinformatic analysis of proteomic and transcriptomic data comparing senescent and non-senescent cells identified several senescent cell anti-apoptotic pathways (SCAPs) [190]. These pathway nodes were targeted with siRNA in senescent versus non-senescent human primary adipocyte progenitors to identify target drugs and natural products. The first adipose senolytic drug that was reported was the above-described D+Q [187]. Clinical use of Dasatinib has been approved since 2006. Quercetin and Fisetin are both natural products that share beneficial safety profiles. All of these compounds can be effectively administered orally, and their elimination half-life is less than 4 h, which implies complete clearance from the human body within 24 h [188].

Approaches are needed to study the effects of the drugs on physiologically produced senescent cells, as well as the effectiveness of the onset of therapy late in life, which is regarded as the most translational use of senolytics. Aging mice may provide one feasible model in this respect, although the use of aged mice is expensive and slow. Therefore, researchers have applied approaches to accelerate aging, including whole-body irradiation, exposure to DNA-damaging agents or a high-fat diet. The Ercc1−/Δ mouse, representing a genetic model of human progeroid syndromes, spontaneously develops senescent cells in multiple tissues because of accumulated endogenous DNA damage, mitochondrial dysfunction, and increased abundance of reactive oxygen species [191]. The fact that naturally aged wild-type mice and progeroid Ercc1−/Δ mice accumulate senescent cells in the same tissues and to the same level makes Ercc1−/Δ mice a good model for studying senolysis [191].

Compared to the mouse, the rat is easier to monitor and generally more reminiscent to that of humans at the physiological level [192]. Indeed, rats are very suitable models for studying human aging and spontaneously develop more “human-like” age-related diseases (e.g., diabetes and sarcopenia) than mice. Indeed, cells derived from different rat tissues have been used to study cellular senescence. Since genetic modification technologies in the rat have advanced, it is now possible to create rat models with transgenes that allow the elimination of cells expressing p16Ink (e.g., an INK-ATTAC rat line).

Explants of human adipose tissue have been used to determine the efficacy of senolytics in humans. Researchers have used major omental adipose explants resected during surgery [193] and cultured small pieces in the presence of senolytics. Conditioned media can be harvested for multiple protein analysis, and tissue can be analyzed for senescence markers, using various innovative technologies that improve our understanding of the biology of aging and, thus, in turn, shift the treatment focus upstream from disease treatment to prevention and health promotion, opening up more possibilities for studying molecular targets.

Most used in vitro cell models to determine the efficacy of senolytics are the 3T3-L1 Mouse Cell Line, a well-established pre-adipose cell line [194], and the 3T3-F442A cell line [195], both developed from murine Swiss 3T3 cells [194,195]. 3T3-L1 cells have been extensively used in evaluating the effects of compounds or nutrients on adipogenesis to determine their potential therapeutic application; for a review, see [196] and references therein.

Using an in vitro model of induced senescence in 3T3-L1 cells differentiated into pre-adipocytes and adipocytes and subjected to repeated treatments with hydrogen peroxide (H_2_O_2_) at a sub-lethal concentration (150 μM), thus inducing a senescent phenotype, Zoico et al. [197] showed that quercetin (20 uM) decreased the number of senescent cells, suppressed the ROS and inflammatory cytokines, and also decreased miR-155-5p [197]. 

Despite the important contribution of in vitro experiments in determining and characterizing the biological actions exerted by various compounds on cellular senescence in adipocytes, they cannot mimic the external environment of cells in tissues of an intact organism. Therefore, in vivo and ex vivo studies are necessary to extend in vitro studies to human testing using clinical trials. In particular, the results of in vivo experiments may provide otherwise inaccessible insights into the mechanisms of senescence induction and facilitate the identification of possible therapeutic targets. It is important to consider that care must be taken to verify whether the chosen model is suitable for the intended experiments and their objectives, as different mouse models and senescence induction methods give different results; for a review, see [198] and the references therein. 

## 8. Conclusions and Future Perspectives

Overall, the evidence presented in this review shows that the dysfunction of adipose mitochondria has a close relationship with cellular senescence and aging-associated metabolic diseases. Counteracting adipose cellular senescence in animal models and humans improves many conditions associated with aging, including insulin resistance, hyperglycemia, and cardiovascular conditions. Several mitochondrial pathways are emerging as promising targets for age-related diseases, and candidate compounds are being developed. 

However, further research, as well as technological innovation, is needed for the in-depth phenotyping of senescent adipose cells. Genomics, epigenomics, proteomics, lipidomic, and metabolomics approaches appear to be useful tools to better define the physiopathological role of mitochondria in the complicated processes related to senescence and to identify new targets to increase the possibilities for healthy aging, with innumerable social and economic benefits.

## Figures and Tables

**Figure 1 ijms-24-11676-f001:**
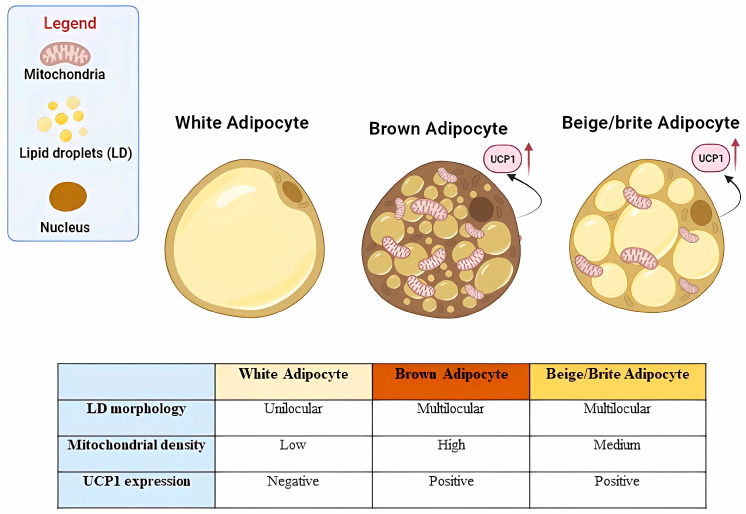
Morphological aspects of white, brown, and beige adipocytes. The white adipocyte, negative to UCP1, displays a single lipid droplet and has a low mitochondrial density. The brown adipocyte, positive to UCP1, is usually smaller than the white adipocyte, presenting a high mitochondrial density and containing multiple small lipid droplets. The beige/brite adipocyte, positive to UCP1, is usually smaller than the white adipocyte, presenting a medium mitochondrial density and containing multiple small lipid droplets. UCP1, uncoupling protein 1; LD, lipid droplet. Created using software from Biorender.com (https://www.biorender.com/, accessed on 16 May 2022).

**Figure 2 ijms-24-11676-f002:**
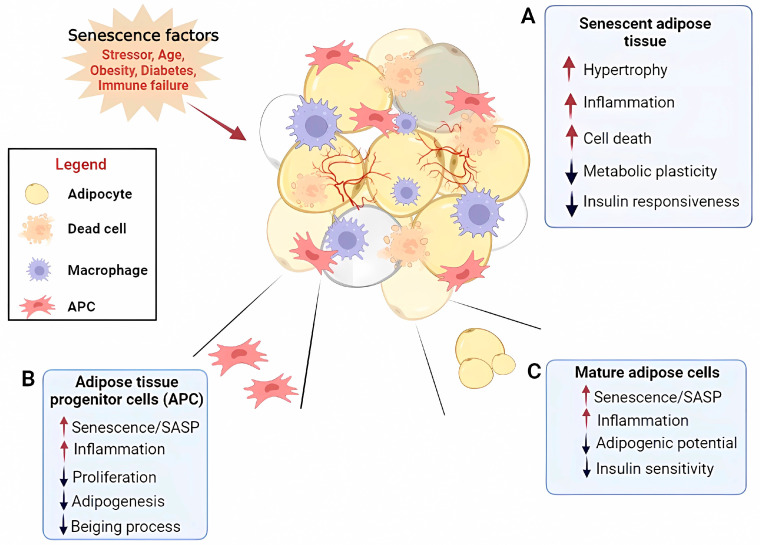
Effects of senescence factors on adipose tissue and its resident cells affecting tissue metabolic plasticity. (**A**) Senescent adipose tissue displays hypertrophy, inflammation, and cell death and is metabolically inflexible (i.e., decreased plasticity and insulin responsiveness). (**B**) Adipose tissue progenitor cells (APCs) display increased SASP and inflammation and decreased proliferation, adipogenesis, and browning processes. (**C**) Mature adipose cells display increased SASP and inflammation and a decreased adipogenicity and insulin sensitivity. SASP, senescence-associated secretory phenotype. Created using software from Biorender.com (https://www.biorender.com/, accessed on 16 May 2022).

**Figure 3 ijms-24-11676-f003:**
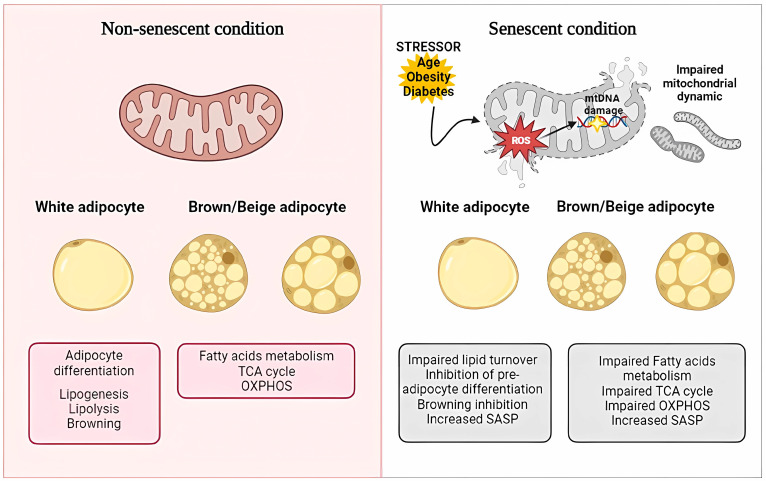
Physiological functions of adipose mitochondria and common impaired responses in stressor-induced senescent conditions. The left panel (in pink) depicts the main functions of mitochondria from white and brown/beige adipocytes in physiological conditions. The right panel (in white) refers to impaired functions of mitochondria from white and brown/beige adipocytes in senescent cells. SASP, senescence-associated secretory phenotype; OXPHOS, oxidative phosphorylation activity; TCA cycle, tricarboxylic acid cycle. Created using software from Biorender.com (https://www.biorender.com/, accessed on 16 May 2022).

**Figure 4 ijms-24-11676-f004:**
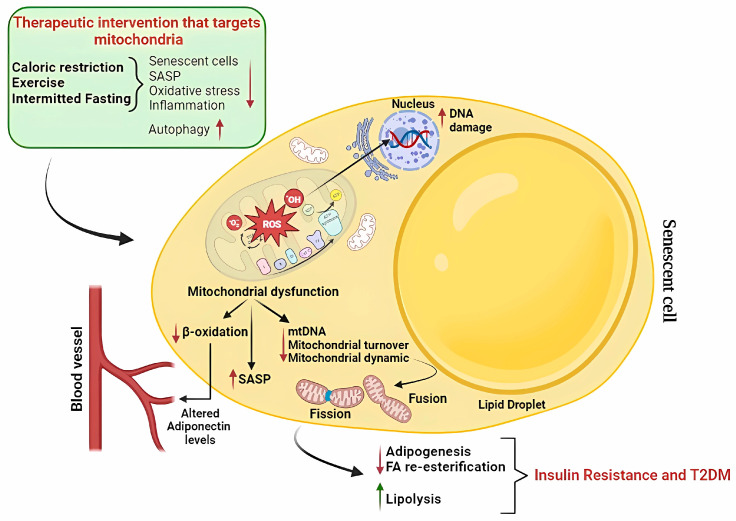
Therapeutic interventions targeting mitochondria in order to tackle metabolic disorders. Mitochondrial dysfunction due to altered mitochondrial biogenesis and ROS-induced oxidative stress affects adipocyte physiology. Lifestyle factors acting on mitochondria attenuate mitochondrial oxidative damage; reduce inflammation, cellular senescence, and SASP; increase autophagy; and modulate adipocyte function, thus providing therapeutic options for the treatment of metabolic disorders. ROS, reactive oxygen species; SASP, senescence-associated secretory phenotype; FA, fatty acids; mtDNA, mitochondrial DNS; T2DM, type 2 diabetes. Created using software from Biorender.com (https://www.biorender.com/, accessed on 16 May 2022).

## Data Availability

Not applicable.

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
