# Peer review of "Physiological Approaches Targeting Cellular and Mitochondrial Pathways Underlying Adipose Organ Senescence"

_ijms, 2023, doi:10.3390/ijms241411676_

Round 1

Reviewer 1 Report

This is a good review article from a well-respected group in the related research filed. The topic addressed is interesting and deserves a constructive discussion.

I would like to make one point: In Fig. 2, there is a notation "dead cell". However, there is no mention of cell death in the text. It would be better if a description of cell death were added.

As a final check, please check the proofreading of the English text.

Author Response

We thank the reviewer for his/her comments.

“I would like to make one point: In Fig. 2, there is a notation "dead cell". However, there is no mention of cell death in the text. It would be better if a description of cell death were added..”

In accordance with the reviewer’s suggestion, a description of cell death was added, see lines 215-220 of the revised version. The requested revision have been  highlighted

English has been checked.

Reviewer 2 Report

The review by de Lange et al. provides important information and discusses that the dysfunction of mitochondria from the adipose tissue plays an important role during senescence and aging-metabolic diseases. Importantly, the authors give valuable information about potential targets in the field of senescence mechanisms as a therapeutic strategy to target mitochondria. However, some points have to be improved and explained.

1)    The description of Figure 1 needs to be improved. Also, the figure can be more explained according to the table shown there.

2)    There is an error in line 86, “This review.”

3)    Several definitions need to be included in the manuscript. For example, in line 131: ERK5

4)    The information in the paragraph in lines 110-112 could be supplemented with experimental articles.

5)    There is some grammar that could be improved. For example, in line 159, there is a reference and then a capital letter.

6)    In lines 167-168 the authors use the word “results” trying to associate mitochondrial dysfunction with adipose tissue inflammation. However, only a study is given. More information is needed, or instead, a modification of the description.

7)    Please modify the letter b in IL-1b.

8)    In line 173, interleukins are not defined until later in line 216.

9)    Please modify the paragraph from lines 261-264 to better clarify the information that was written.

10) The description of Figure 3 does not match the color mentioned in the text compared to the image.

11) Please mention in the text which mechanism or molecular pathway is associated with the markers b-gal, p16, and PAI1 (line 361).

12) Please carefully check abbreviations along the text.

13) It is suggested to separate into sections the therapeutic approaches for cellular senescence. For example Caloric restriction, exercise, intermittent fasting, etc.

14) Abbreviations should be used consistently: -ATP, ARN, TCA, DNA, ETC, PPARg, LPS, ROS, HFD, FFA, b-GAL should be defined. -SIRT is defined twice -LINE 142: “Brown adipose tissue” should be “BAT”.

15)Figures: The quality of the entire figure should enhance. The letters used are blurred and in some cases are very small (Figs. 2 and 4)

Figure 1: The color of brown adipose is not contrasting.

Legend to Figure 2: SASP should be defined.

Legend to Figure 3: OXPHOS, TCA, SASP and ROS should be defined.

Legend to Figure 4: FA, SASP and ROS should be defined.

16) References 31 and 46 are not cited in the text.

Author Response

We thank the reviewer for his/her constructive comments that improved the review. We answered to all the comments as reported in the following point-by-point response. All the revisions have been highlighted in the revised version.

“ The description of Figure 1 needs to be improved. Also, the figure can be more explained according to the table shown there”.

In accordance with the reviewer, we have improved Figure 1.

“ There is an error in line 86, “This review.””

The error has been corrected.

“ Several definitions need to be included in the manuscript. For example, in line 131: ERK5”

Definitions have been included along  the manuscript and highlighted in red.

“The information in the paragraph in lines 110-112 could be supplemented with experimental articles”

Experimental articles have been supplemented, see references 24 and 25 and the text lines 111-115 of the revised version.

“ There is some grammar that could be improved. For example, in line 159, there is a reference and then a capital letter”.

The grammar has been checked throughout the MS.

“ In lines 167-168 the authors use the word “results” trying to  associate mitochondrial dysfunction with adipose tissue inflammation. However, only a study is given. More information is needed, or instead, a modification of the description”.

In accordance with the reviewer, the description has been modified, see lines 192-196 of the revised version.

“Please modify the letter b in IL-1b”

The letter has been modified, see line 200 of the revised version.

“  In line 173, interleukins are not defined until later in line 216”.

Interleukins have been defined.

“Please modify the paragraph from lines 261-264 to better clarify the information that was written”

In accordance with the reviewer the description has been modified, see lines 289-293 of the revised version.

“The description of Figure 3 does not match the color mentioned in the text compared to the image”

Correction has been made. See the revised fig. 3

“Please mention in the text which mechanism or molecular pathway is associated with the markers b-gal, p16, and PAI1 (line 361)”.

In accordance with the reviewer, the mechanisms has been mentioned, see lines 419-425 of the revised version.

“ Please carefully check abbreviations along the text”.

Abbreviations have been checked along the text.

“ It is suggested to separate into sections the therapeutic approaches for cellular senescence. For example Caloric restriction, exercise, intermittent fasting, etc”.

In accordance with the reviewer’s suggestion, we now arranged the section  into paragraphs with headings.

“Abbreviations should be used consistently: -ATP, ARN, TCA, DNA, ETC, PPARg, LPS, ROS, HFD, FFA, b-GAL should be defined. -SIRT is defined twice -LINE 142: “Brown adipose tissue” should be “BAT”.

Abbreviations have been checked along the text.

“Figures: The quality of the entire figure should enhance. The letters used are blurred and in some cases are very small (Figs. 2 and 4)” Figure 1: The color of brown adipose is not contrasting”

 The Figures have been enhanced in quality and better described.

“Legend to Figure 2: SASP should be defined. Legend to Figure 3: OXPHOS, TCA, SASP and ROS should be defined. Legend to Figure 4: FA, SASP and ROS should be defined.

 Abbreviations have been defined in legends to figure.

References 31 and 46 are not cited in the text.

References are now cited in the text. Moreover, new references have been added due to the implementation of the various section along the manuscript.

Reviewer 3 Report

This presented manuscript by de Lange and coauthors, aims to describe present the current state of the mitochondria impact on adipocytes senescence. The text is easy to follow and can be potentially interesting for the readers, but it needs to be improved before being accepted for publication.

If possible, the answers and comments should be included in a new version of manuscript, since they will allow simpler following of presented data.

Line 86- should be “is”

Lines 111-112 –  I suggest the rewrite the sentence since it is not easy to follow;

Information presented in lines 110-131 are too general; add more information about FOXO3 involvement in adipose tissue activity, as well as about SIRT1, present adipogenic and anti-adipogenic factors, miRNAs.

Lines 132-142 – the issue presented here very important, therefore present more details about the type of thermogenic markers detected;

Line 152 – present the full name of “p38/MAPK-p16Ink4a”; present more information about this pathway;

Lines 163-165 – the sentence needs to be rewritten;

Line 167 – present more details about effect of complex IV disruption on adipocyte activity;

Line 172 – present full names of cytokines;

Lines 174-175 – explain the involvement of “p16INK4A”in senescence;

Line 176 – present full names of FFA, LPS; in the text present the source of FFA, LPS suitable to the presented information;

Line 181 -  present full name of NFkB;

Figure 2 – it needs to have more molecular information (at least SASP), which are presented within the text; otherwise it is unnecessary; I suggest to present figure showing factors inducing senescence in adipocytes.

Lines 196-197 – rewrite the sentence; add more details;

Line 197 – explain what means “poorly differentiated human senescent preadipocytes” and how it can be performed (in vitro and in vivo); what are properly differentiated preadipocytes?

Line 228, and within the manuscript- to SASP belongs some cytokines, i.e. TNF-α. In the text it is very often repeated “SASP factors and TNF-α”, etc. “SASP factors such as IL-6, activin A and TNF-α”, etc.

Lines264-265 – metabokines are very important - add more details;

Lines 300-301- in my opinion presented statement needs to be commented more deeply;

Figure 3 – the agents inducing mitochondrial disruptions  are not clearly stated and figure needs to be improved;

Line 329 – sentence needs to be rewritten;

Line 331 – what myokines and when can effect adipocytes - add details;

Lines 335-336 – it should ne TNFa, MCP

Line 339 – what enzyme?

Line 351 – add shortly the features of “skeletal muscle ageing”

Line 356 – present full name of CDKN1A, CDKN2A

Chapter 6 – since this part of manuscript is very important I suggest to give more details of molecular impact of each mentioned factors, especially mentioned in the text for the first time; if it is possible add some doses/concentrations and time of treatment, quantitative effect on some parameters; add some examples of the CR mimetics present in human diet effectively lowering the senescence of adipocytes; present the effect of exercise as a new paragraph; are there any known epigenetic mechanisms involved in adipocytes senescence?

Chapter 7 – add the quantitative data (i.e. concentration/dose, incubation/time of treatment, quantitative effect)

Lines 530-540 – can the Authors compare the models and present their pros and cons? I suggest to add shortly some in vitro cell lines models with details allowing the induction of senescence.

In summary, the manuscript requires the major revision.

Author Response

We thank the reviewer for his/her constructive comments that improved the review.We answered to all the comments as reported in the following point-by-point response. All the revisions have been highlighted in the revised version.

"Line 86- should be “is”"

The correction has been made.

"Lines 111-112 –  I suggest the rewrite the sentence since it is not easy to follow; Information presented in lines 110-131 are too general; add more information about FOXO3 involvement in adipose tissue activity, as well as about SIRT1, present adipogenic and anti-adipogenic factors, miRNAs."

In accordance with the reviewer, we have rewritten the paragraph to be more clear and complete in the description. See lines 111-128 of the revised version.

"Lines 132-142 – the issue presented here very important, therefore present more details about the type of thermogenic markers detected"

In accordance with the reviewer, we have presented more details  in the description. See lines 166-183 of the revised version.

"Line 152 – present the full name of “p38/MAPK-p16Ink4a”; present more information about this pathway"

In accordance with the reviewer, we have presented the full name and gave more information about the pathway.  See lines 155-165 of the revised version.

"Lines 163-165 – the sentence needs to be rewritten"

In accordance with the reviewer, we have rewritten the sentence.  See lines 186-189 of the revised version.

"Line 167 – present more details about effect of complex IV disruption on adipocyte activity"

In accordance with the reviewer, more details have been added. See lines 192-196 of the revised version.

"Line 172 – present full names of cytokines"

Full names have been presented.

"Lines 174-175 – explain the involvement of “p16INK4A”in senescence"

In accordance with the reviewer, we have presented more details  in the description. See lines 197-212 of the revised version

"Line 176 – present full names of FFA, LPS; in the text present the source of FFA, LPS suitable to the presented information; Line 181 -  present full name of NFkB"

Full names have been presented along the manuscript

"Figure 2 – it needs to have more molecular information (at least SASP), which are presented within the text; otherwise it is unnecessary; I suggest to present figure showing factors inducing senescence in adipocytes".

In accordance with the reviewer, Figure 2 has been integrated with factors inducing senescence in adipocytes.

"Lines 196-197 – rewrite the sentence; add more details; Line 197 – explain what means “poorly differentiated human senescent preadipocytes” and how it can be performed (in vitro and in vivo); what are properly differentiated preadipocytes?"

In accordance with the reviewer, we have presented more details  in the description. See lines 221-233 of the revised version.

"Line 228, and within the manuscript- to SASP belongs some cytokines, i.e. TNF-α. In the text it is very often repeated “SASP factors and TNF-α”, etc. “SASP factors such as IL-6, activin A and TNF-α”, etc."

Corrections have been made.

"Lines264-265 – metabokines are very important - add more details";

In accordance with the reviewer, we have presented more details  in the description. See lines 289-310 of the revised version.

Lines 300-301- in my opinion presented statement needs to be commented more deeply;

In accordance with the reviewer, we have presented more details  in the description. See lines 336-350 of the revised version.

"Figure 3 – the agents inducing mitochondrial disruptions are not clearly stated and figure needs to be improved"

The figure has been modified.

"Line 329 – sentence needs to be rewritten"

In accordance with the reviewer, the sentence has been rewritten. See lines 371-376 of the revised version.

"Line 331 – what myokines and when can effect adipocytes - add details";

Details have been added, See lines 384-391 of the revised version.

"Lines 335-336 – it should ne TNFa, MCP. Line 339 – what enzyme?"

Corrections have been done.

"Line 351 – add shortly the features of “skeletal muscle ageing”"

In accordance with the reviewer, we have shortly added the features of “skeletal muscle ageing”. See lines 393-411 of the revised version.

"Line 356 – present full name of CDKN1A, CDKN2A"

Full names have been presented.

"Chapter 6 – since this part of manuscript is very important I suggest to give more details of molecular impact of each mentioned factors, especially mentioned in the text for the first time; if it is possible add some doses/concentrations and time of treatment, quantitative effect on some parameters; add some examples of the CR mimetics present in human diet effectively lowering the senescence of adipocytes; present the effect of exercise as a new paragraph; are there any known epigenetic mechanisms involved in adipocytes senescence?"

To meet the reviewer suggestions, we have changed the chapter 6,  presented more details  in the description and added some examples of the CR mimetics present in human diet effectively lowering the senescence of adipocytes as well as some epigenetic mechanisms involved in adipocytes senescence. We now arranged the section  into paragraphs with headings. See lines 444-450 and 482-489 495-512 518-526 of the revised version.

"Chapter 7 – add the quantitative data (i.e. concentration/dose, incubation/time of treatment, quantitative effect)" "Lines 530-540 – can the Authors compare the models and present their pros and cons? I suggest to add shortly some in vitro cell lines models with details allowing the induction of senescence.

To meet the reviewer suggestions, we have changed the chapter 7, added quantitative data and some in vitro cell lines models with details allowing the induction of senescence. See lines 625-645 and 690-713 of the revised version.

New references have been added due to the implementation of the various section along the manuscript.

Round 2

Reviewer 3 Report

I have read the Authors’ response and they answered to my concerns.  The submitted manuscript has been significantly improved. In my opinion, the manuscript can be accepted for publication in IJMS journal. I have only a one comment – I suppose that “Therapeutic approaches for cellular senescence” part should be numbered as “6”.